# Younger and rural children are more likely to be hospitalized for SARS-CoV-2 infections

Rebecca M. Cantu[1,2]*, Sara C. Sanders[1,2], Grace A. Turner[3], Jessica N. Snowden[2,3,4], Ashton Ingold[3], Susanna Hartzell[3], Suzanne House[3,5], Dana Frederick[3,5], Uday K. Chalwadi[6], Eric R. Siegel[7], Joshua L. Kennedy [2,3,5,8]

1 Division of Hospital Medicine, Department of Pediatrics, University of Arkansas for Medical Sciences, Little Rock, Arkansas, United States of America, 2 Arkansas Children's Hospital, Little Rock, Arkansas, United States of America, 3 Arkansas Children's Research Institute, Little Rock, Arkansas, United States of America, 4 Division of Infectious Diseases, Department of Pediatrics, University of Arkansas for Medical Sciences, Little Rock, Arkansas, United States of America, 5 Division of Allergy and Immunology, Department of Pediatrics, University of Arkansas for Medical Sciences, Little Rock, Arkansas, United States of America, 6 Department of Pediatrics, University of Arkansas for Medical Sciences, Little Rock, Arkansas, United States of America, 7 Department of Biostatistics, University of Arkansas for Medical Sciences, Little Rock, Arkansas, United States of America, 8 Division of Pulmonary and Critical Care Medicine, University of Arkansas for Medical Sciences Department of Internal Medicine, Little Rock, Arkansas, United States of America

* rmcantu@uams.edu

**Data Availability Statement:** Because of the confidential nature of some of the data and our institutional review board (IRB) restrictions, data will only be provided upon request and with an

## Abstract

### Purpose

To identify characteristics of SARS-CoV-2 infection that are associated with hospitalization in children initially evaluated in a Pediatric Emergency Department (ED).

### Methods

We identified cases of SARS-CoV-2 positive patients seen in the Arkansas Children's Hospital (ACH) ED or hospitalized between May 27, 2020, and April 28, 2022, using ICD-10 codes within the Pediatric Hospital Information System (PHIS) Database. We compared infection waves for differences in patient characteristics and used logistic regressions to examine which features led to a higher chance of hospitalization.

### Findings

We included 681 pre-Delta cases, 673 Delta cases, and 970 Omicron cases. Almost 17% of patients were admitted to the hospital. Compared to Omicron-infected children, pre-Delta and Delta-infected children were twice as likely hospitalized (OR = 2.2 and 2.0, respectively; $p<0.0001$). Infants under one year were >3 times as likely to be hospitalized than children ages 5–14 years regardless of wave (OR = 3.42; 95%CI = 2.36–4.94). Rural children were almost three times as likely than urban children to be hospitalized across all waves (OR = 2.73; 95%CI = 1.97–3.78). Finally, those with a complex condition had nearly a 15-fold increase in odds of admission (OR = 14.6; 95%CI = 10.6–20.0).

approved IRB protocol and Data Utilization
Agreement. Data requests can be sent to Jill
Gassaway, Director of Research Integrity at ACRI
via gassawayja@ararchildrens.org.

**Funding:** Dr. Joshua Kennedy, Mr. Eric Siegel, Ms.
Grace Turner, Ms. Ashton Ingold, Ms. Susanna
Hartzell, Ms. Suzanne House, and Ms. Dana
Frederick are supported by the Center for
Translational Pediatric Research (NIH/NIGMS
P20GM121293, Team Science Supplement and
SARS-CoV-2 Variant Sequencing Supplement). In
addition, Drs. Joshua Kennedy, Uday Chalwadi,
and Mr. Eric Siegel are supported by the
Translational Research Institute (NIH/NCATS
UL1TR003107). The funding organizations had no
roles in the study design, data collection and
analysis, decision to publish, or preparation of the
manuscript.

**Competing interests:** Dr. Kennedy has been a
consultant for Genentech for asthma. Funding
organizations were not involved in the design or
conduct of the study, nor were they engaged in the
collection, analysis, or interpretation of the data,
nor were they involved in the preparation, editing,
or censuring of the manuscript.

## Conclusions

Children diagnosed during the pre-Delta or Delta waves were more likely to be hospitalized than those diagnosed during the Omicron wave. Younger and rural patients were more likely to be hospitalized regardless of the wave. We suspect lower vaccination rates and larger distances from medical care influenced higher hospitalization rates.

## Background

COVID-19 was first reported in late December of 2019 in Wuhan, China, with the first case in the United States reported on January 18, 2020, in Washington state [1]. As of August 2022, SARS-CoV-2 has infected over 600 million individuals worldwide and caused over 6 million deaths. Over two years, SARS-CoV-2 has undergone numerous genetic mutations, resulting in multiple variant strains, each capable of different transmission rates, risks for severe disease, and even risks for mortality [2]. The Delta variant, also known as B.1.617.2, was shown to be 40%-60% more transmissible than the Alpha variant [3]. Studies have determined that Delta also increased hospitalization risk by 108%, increased intensive care unit (ICU) admission by 235%, and caused a 133% higher mortality in adults compared to Alpha [4]. The Omicron variant, B.1.1.529, became the predominant strain in the United States by December 2021. Soon after, scientists discovered that though symptoms of this variant were seemingly less severe, Omicron was more transmissible than Delta and had increased resistance to antiviral immunity [5–7].

Much of the research on SARS-CoV-2 and its variants has focused on adults, yet the effect of variants in children has been largely ignored. This is likely because, early in the pandemic, pediatric COVID-19 infections presented with milder symptoms than adult infections. However, recent data have shown that, since December 2021, hospitalization rates increased more rapidly in children < 5 years than in any other age group [8]. The rise of the Omicron variant significantly increased the number of pediatric cases, from < 2% of total reported cases in the early pandemic to 25% of U.S. cases by early February 2022 [9].

Many studies that focused on hospitalized children with SARS-CoV-2 infection were either done early in the pandemic (pre-Omicron) when the most significant risk was older age and co-morbidities [10, 11] or focused upon larger, more populated regions, where there were more children with SARS-CoV-2 to study [8, 12, 13]. We previously reported on demographic and clinical factors associated with pediatric hospitalizations in Arkansas from March 2020 to December 2020 [14]. In this study, rural children were more likely to be hospitalized during SARS-CoV-2 infection. Given the increased complexity of patients seen when the Delta variant was rampant and the explosion of infections when the Omicron variant predominated, we wished to extend this study and determine the effect of the newer variants. With this in mind, we aimed to identify characteristics of SARS-CoV-2 infection associated with hospitalization in children initially evaluated in a Pediatric Emergency Department from a largely rural state. We hypothesized that specific demographic and clinical factors would be related to hospitalization in a pediatric population infected with SARS-CoV-2. We also evaluated whether these differences were variant-specific within our population.

## Methods

This cross-sectional study utilized the Pediatric Hospital Information System (PHIS) Database queried on July 7, 2022. PHIS is a pediatric database with clinical and resource utilization data for more than 50 children's hospitals, including Arkansas Children's Hospital (ACH). This

study was approved by the University of Arkansas for Medical Sciences Institutional Review Board (IRB #23497) with a waiver of consent and HIPAA authorization.

This study included children ≤18 years of age who presented to ACH between May 27, 2020, and April 28, 2022. Inclusion criteria for this study were emergency department (ED) visit or hospital admission during the study visit and COVID or related diagnoses as defined by ICD-10 codes, including U07.1 (COVID), M35.81 (Multisystem Inflammatory Syndrome in Children, MIS-C), and J12.82 (Pneumonia due to coronavirus disease 2019). In addition, those with diagnoses of other systematic connective tissue involvement (either "other specified' or "unspecified," ICD-10 codes M35.8 or M35.9, respectively), two codes utilized for MIS-C before the release of the final ICD-10 code [15], were combined with the MIS-C group.

For this study, we focused on children seen in the Emergency Department (ED) at ACH so that we could validate information received via the PHIS database in our home institution's electronic health record. To confirm and validate the ICD-10 diagnosis codes, we randomly selected 241 (10%) charts for review. All 241 charts had these diagnosis codes associated with an ED and/or hospital visit, and ~85% had a positive viral test in the medical record at the time of the visit. PHIS reports were also cross-referenced for accuracy with data from the ACH Infection Prevention Department, which maintains a list of patients identified as having COVID-19, and a list from the University of Arkansas for Medical Sciences Department of Pediatrics Section of Infectious Diseases of patients with MIS-C.

Data elements collected from the PHIS Database included age, sex, race/ethnicity based on the Equity Race Category (ERC) classifications provided by PHIS, rurality based on rural-urban commuting-area (RUCA) code, visit type (ED visit, inpatient, or observation), intensive care unit (ICU) admission, length of stay (LOS), presence of a complex chronic-condition flag in PHIS, [16] disposition, mortality, and diagnosis of MIS-C. A chart review was conducted to validate the ERC-based race/ethnicity groups for subjects obtained through the PHIS database. After reviewing the ACH medical record, we found that many of those assigned to Race = Other by PHIS data were Hispanic and/or Latino. Reviewed subjects whose ACH information indicated they were Hispanic and/or Latino were deemed to have been misclassified by the PHIS algorithm and were accordingly reclassified as Hispanic.

Cases were assigned to the then-predominant SARS-CoV-2 variant based on sequencing performed at similar times in pediatric samples from Arkansas. Unfortunately, no sequencing was available before March 2021 in the state during what was likely the alpha wave. Therefore, we assigned cases by admission date into the pre-Delta (May 27, 2020-May 1, 2021), Delta (June 1, 2021-November 15, 2021), and Omicron groups (December 16, 2021- March 31, 2022). The month-long transition periods between waves and after Omicron were defined to ensure that a variant of concern would be present in higher numbers and include less overlap between each variant (**Fig 1**). The n = 86 cases that had admission dates falling into one of these transition periods were not included in the statistical analyses, but data for them can be found in **S1 (demographics) and S2 (disease characteristics) Tables.**

## Data preprocessing

To combine multiple records from the same case, we defined an admission as a unique combination of medical record number (MRN), admission date, and discharge date. Two or more records from the same admission were combined to preserve the "Yes" value of any Yes/No "flag" variable that changed between records. If the same MRN had a second admission ≤30 days after the discharge date from their first admission, this was considered a readmission of the same case. Records from two or more admissions of the same case were combined as described above while setting the admission date, discharge date, and age at admission to be

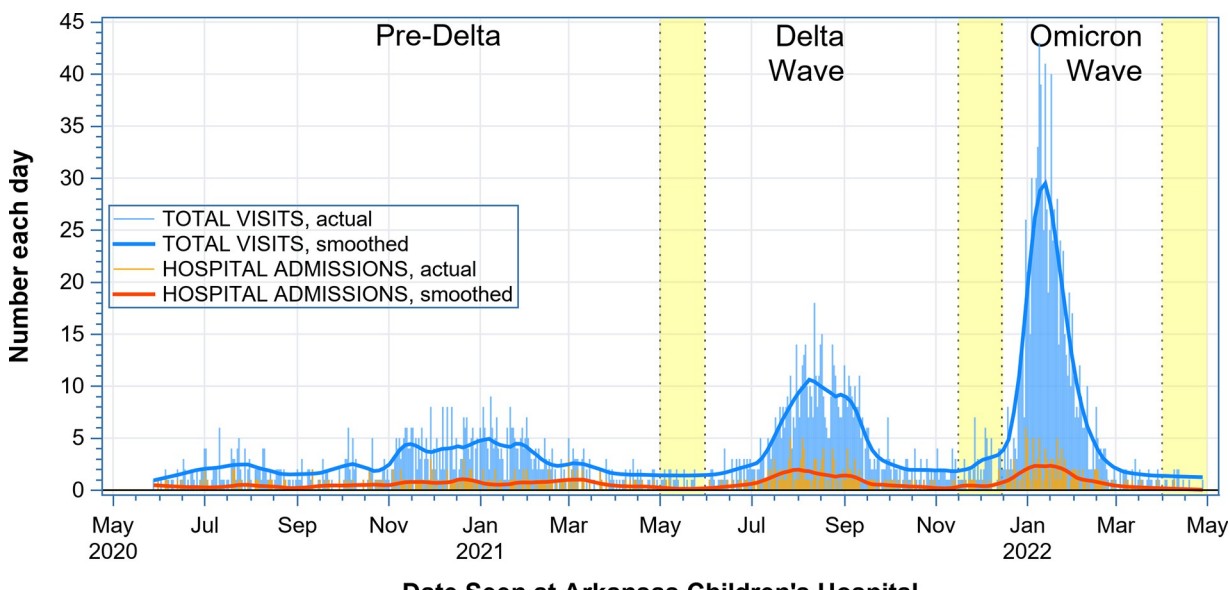

**Fig 1. Total visits (ED and hospitalizations) and hospitalization data for ACH.** Spikes (labeled "actual" in the legend) show the actual numbers of total visits (lighter blue) and hospital admissions (lighter orange) seen at ACH on each calendar day of the study period. Heavier curves (labeled "smoothed" in the legend) were generated by loess regressions on the actual numbers and show the locally weighted average numbers of total visits (darker blue) and hospital admissions (reddish orange) throughout the study. The uncolored regions represent the three infection waves attributed to the pre-Delta, Delta, and Omicron SARS-CoV-2 variants. The two yellow regions between waves plus the 3rd yellow region after the Omicron wave denote the 1-month-long transition periods.

the first admission date, last discharge date, and age at first admission. Conversely, if the same MRN had a second admission ≥31 days after the last discharge date from their first admission, this was considered a new case and labeled a second infection. We chose 30 days as our cut-off based on the Centers for Medicare and Medicaid definition for readmissions [14, 17].

## Statistical analysis

Age in years and LOS in days (d) were summarized by groups as means and standard deviations [SDs] as well as medians and quartiles and assessed for the presence of differences among the three COVID groups with the Kruskal-Wallis test. In addition, age groups, other categorical variables, and flag variables were compared for differences between pre-Delta, Delta, and Omicron groups with the Pearson chi-square test. To analyze Race/Ethnicity, we retained as-is the three ERC classifications of Hispanic, Non-Hispanic Black, and Non-Hispanic White that frequently occurred in our data extract but condensed the four infrequent ERC classifications of Asian, Multiracial, Other, and Unknown into a single group named "Other." We employed logistic regression in univariable and multivariable modes after excluding 22 cases missing RUCA codes to examine the independent associations of patient characteristics with hospital admission. The seven predictors used in both modes were Infection Wave, Sex, Age Group, ERC-based Race/Ethnicity, RUCA-based Urban status, Payor Type, and Complex Chronic Condition. All seven predictors were entered without variable selection in the multivariable logistic regression model, which contained no interaction terms. For subgroup analyses of hospital admission within each infection wave, we repeated this multivariable logistic regression procedure for the six remaining predictors. All statistical hypothesis tests employed a 2-sided $\alpha = 0.05$ significance level, and all analyses were performed using SAS v9.4 software (SAS Institute Inc., Cary, NC, USA).

## Results

### Cohort characteristics

We identified 2410 individual cases of SARS-COV-2-positive patients seen in the ED or admitted to observation or inpatient at ACH between May 27, 2020, and April 28, 2022. Their mean age in years was 5.91 (standard deviation [SD] 6.04), and 27.5% (n = 662) were less than one year of age. Forty-seven percent (n = 1141) were female, and 83.8% (n = 2019) lived in an urban area code. The proportion of subjects admitted to the hospital was 16.6% (n = 400). Thirteen percent (n = 303) had complex chronic conditions. The 2410 cases included 681 during pre-Delta, 673 during Delta, and 970 during Omicron. Table 1 provides patient demographics.

### Primary comparisons of demographic and clinical characteristics

Concerning admissions, we found differences between the variant waves. During the pre-Delta wave, 142 pediatric patients (20.9%) were admitted to the hospital. 130 patients (19.3%) were admitted during the Delta wave, and 116 (12%) were admitted during the Omicron wave. Table 1 shows that race varied significantly between waves (p<0.001) in a complex manner. The proportion of non-Hispanic Black patients fell slightly with each successive wave, from 34.7% pre-Delta to 30.8% during Delta and 26.6% during Omicron. In contrast, the proportion of Hispanic patients fell noticeably from 27.8% pre-Delta to 16.5% during Delta but rebounded to 25% (n = 242) of the total population during Omicron. In partial compensation, the proportion of White subjects increased noticeably from 32.9% pre-Delta to 43.2% during Delta before falling to 39.6% during Omicron. In comparison, the proportion of Other rose from 4.7% pre-Delta to 9.5% during Delta and 8.9% during Omicron (Table 1). Between each variant wave, there were differences in the ages of infected subjects. In subjects four years of age and less, the proportion of patients infected with SARS-COV-2 stayed nearly equal from pre-Delta (46.6%) to Delta (47.5%) but then jumped more than 15 percentage points (to 62.9%) during Omicron (Table 1). In comparison, those 5–18 years old decreased in mirror fashion each period. Rates of MIS-C reached 4.7% (n = 32/681) in the pre-Delta wave, compared to 1.9% (n = 13/673) in the Delta wave and 0.6% (n = 6/970) in the Omicron wave. LOS in days was significantly different (p<0.001) between children hospitalized for SARS-CoV-2 infections between variant waves. Those hospitalized during the pre-Delta wave had a mean LOS of 0.86d (standard deviation [SD], 2.30d). In contrast, those hospitalized during Omicron had a mean LOS of 0.46d (SD 1.79d), and those hospitalized during the Delta wave had a mean LOS of 1.36d (SD 6.91d). None of the other demographic factors showed a statistically significant change between COVID variants (Table 1).

### Univariate and multivariate logistic regression for all waves

Univariate and multivariate logistic regressions were employed to assess risk factors for hospitalization across the entire study; Table 2 shows the results. The unadjusted and adjusted odds ratios (ORs) show that the odds of hospitalization were highest during the pre-Delta wave and declined only slightly during the Delta wave, compared to the Omicron wave as reference (Table 2). Age group was not a significant risk factor in univariate analysis but became a critical risk factor in multivariate analysis. In particular, the OR of admission (95% confidence interval) for infants <1 year old increased from 1.23 (0.92–1.62) before multivariate adjustment to 2.35 (1.67–3.29) after multivariate adjustment, whereas the ORs of admission for the other age groups changed only modestly (Table 2). In contrast, sex, race/ethnicity, and payor type were univariately significant risk factors that lost their significance after multivariate

**Table 1. Demographics.**

| Demographic Factor (Whole study) | Whole Study (N = 2,410) | Pre-Delta Wave (N = 681) | Delta Wave (N = 673) | Omicron Wave (N = 970) | P value[†] |
|---|---|---|---|---|---|
| **Age in years** | | | | | **<0.001**[§] |
| Mean [SD[1]] | 5.91 [6.04] | 6.99 [6.39] | 6.60 [6.0] | 4.72 [5.56] | |
| Median (1st– 3rd Quartiles) | 3 (0–11) | 6 (1–13) | 5 (1–12) | 2 (0–9) | |
| **CDC Age Group, % (N)** | | | | | **<0.001** |
| Under 1 year of age | 27.5% (662) | 24.4% (166) | 23.6% (159) | 32.2% (312) | |
| 1–4 years of age | 26.2% (632) | 22.0% (150) | 23.9% (161) | 30.7% (298) | |
| 5–14 years of age | 32.0% (771) | 34.4% (234) | 36.4% (245) | 27.0% (262) | |
| 15–18 years of age | 14.3% (345) | 19.2% (131) | 16.0% (108) | 10.1% (98) | |
| **Sex, % (N)** | | | | | 0.498 |
| Female | 47.3% (1,141) | 46.3% (315) | 49.2% (331) | 46.7% (453) | |
| Male | 52.7% (1,269) | 53.7% (366) | 50.8% (342) | 53.3% (517) | |
| **Race[2], % (N)** | | | | | **<0.001** |
| Hispanic | 23.3% (561) | 27.8% (189) | 16.5% (111) | 25.0% (242) | |
| Non-Hispanic Black | 29.8% (717) | 34.7% (236) | 30.8% (207) | 26.6% (258) | |
| Non-Hispanic White | 39.1% (942) | 32.9% (224) | 43.2% (291) | 39.6% (384) | |
| Other | 7.9% (190) | 4.7% (32) | 9.5% (64) | 8.9% (86) | |
| **Urban[3], % (N)** | | | | | 0.095 |
| Urban | 83.8% (2,019) | 84.0% (572) | 83.1% (559) | 83.8% (813) | |
| Non-Urban | 13.6% (328) | 13.1% (89) | 15.5% (104) | 12.8% (124) | |
| Unknown | 2.6% (63) | 2.9% (20) | 1.5% (10) | 3.4% (33) | |
| **Admitted, % (N)** | 16.6% (400) | 20.9% (142) | 19.3% (130) | 12.0% (116) | **<0.001** |
| **Any Complex Care Code[4], % (N)** | 12.6% (303) | 12.6% (86) | 13.4% (90) | 12.1% (117) | 0.733 |
| **Principal Diagnosis, % (N)** | | | | | **< .001** |
| COVID-19 Only | 97.6% (2,352) | 95.0% (647) | 97.5% (656) | 99.4% (964) | |
| More Serious | 2.4% (58) | 5.0% (34) | 2.5% (17) | 0.6% (6) | |
| *MIS-C* | *2.1% (51)* | *4.7% (32)* | *1.9% (13)* | *0.6% (6)* | |
| *COVID-19 Pneumonia* | *0.3% (7)* | *0.3% (2)* | *0.6% (4)* | *0% (0)* | |
| **ICU Use?, % (N)** | 3.0% (73) | 4.3% (29) | 3.7% (25) | 1.6% (16) | **0.004** |
| **Length of Stay, days (mean [SD[1]])** | 0.83 [4.04] | 0.86 [2.30] | 1.36 [6.91] | 0.46 [1.79] | **<0.001**[§] |

†: *P* values are from either Pearson chi-square or

§Kruskal-Wallis tests. **Bold** denotes significance at α = 0.05 (2-sided).

1: Standard Deviation.

2: Based on Equity Race Category classifications.

3: Based on Rural-Urban Commuting-Area (RUCA) Code.

4: Any Complex Chronic Condition.

adjustment (**Table 2**). In addition, if the patient was considered rural based on their home zip code, their odds of admission were roughly 3-fold (3x) higher than patients from urban zip codes in both univariate and multivariate analysis (**Table 2**). Finally, patients with a complex chronic condition had >10x higher odds of admission when compared to those without in both univariate and multivariate analysis (**Table 2**).

## Univariate and multivariate logistic regression between waves

Next, we performed multivariate logistic regressions subgrouped by the variant wave into which the patient was classified; **Fig 2** shows the resulting adjusted ORs for hospitalization. In

**Table 2. Risk of hospitalization in SARS-COV-2-infected pediatric patients.**

| Outcome: Any Hospitalization | Univariate Analyses | | | | Multivariate Analysis | | | |
|---|---|---|---|---|---|---|---|---|
| | Unadjusted Odds Ratio | 95% Conf. Limits | | p-value | Adjusted Odds Ratio | 95% Conf. Limits | | p-value |
| | | Lower | Upper | | | Lower | Upper | |
| **Infection Wave** | | | | < .0001 | | | | < .0001 |
| Pre-Delta | 1.940 | 1.484 | 2.536 | | 2.511 | 1.831 | 3.444 | |
| Delta | 1.763 | 1.342 | 2.315 | | 2.069 | 1.508 | 2.840 | |
| Omicron[†] | 1.00 | — | — | | 1.00 | — | — | |
| **Sex** | | | | 0.0398 | | | | 0.3008 |
| Male | 1.260 | 1.011 | 1.570 | | 1.143 | 0.887 | 1.473 | |
| Female[†] | 1.00 | — | — | | 1.00 | — | — | |
| **Age Group** | | | | 0.1320 | | | | < .0001 |
| <1 year old | 1.225 | 0.924 | 1.623 | | 2.345 | 1.673 | 3.287 | |
| 1–4 years old | 0.922 | 0.683 | 1.244 | | 1.379 | 0.968 | 1.964 | |
| 5–14 years old[†] | 1.00 | — | — | | 1.00 | — | — | |
| 15–18 years old | 1.288 | 0.921 | 1.801 | | 1.228 | 0.821 | 1.836 | |
| **Race/Ethnicity** | | | | .0005 | | | | 0.4008 |
| Hispanic | 0.533 | 0.393 | 0.724 | | 0.742 | 0.515 | 1.069 | |
| Non-Hispanic Black | 0.760 | 0.587 | 0.984 | | 0.893 | 0.652 | 1.224 | |
| Other | 0.652 | 0.418 | 1.018 | | 0.782 | 0.469 | 1.305 | |
| Non-Hispanic White[†] | 1.00 | — | — | | 1.00 | — | — | |
| **Payor Type** | | | | 0.0001 | | | | 0.1712 |
| Commercial | 1.769 | 1.356 | 2.306 | | 1.352 | 0.977 | 1.871 | |
| Other/Self/Unknown | 1.033 | 0.644 | 1.658 | | 0.951 | 0.550 | 1.646 | |
| Medicaid[†] | 1.00 | — | — | | 1.00 | — | — | |
| **Urban** | | | | < .0001 | | | | < .0001 |
| No | 3.589 | 2.758 | 4.672 | | 2.757 | 2.016 | 3.770 | |
| unknown | 1.352 | 0.697 | 2.625 | | 1.135 | 0.531 | 2.428 | |
| Yes[†] | 1.00 | — | — | | 1.00 | — | — | |
| **Complex Chronic Condition** | | | | < .0001 | | | | < .0001 |
| Yes | 12.415 | 9.451 | 16.309 | | 14.075 | 10.404 | 19.043 | |
| No[†] | 1.00 | — | — | | 1.00 | — | — | |

†Reference category

‡MIS-C or Pneumonia Due To COVID-19.

the pre-Delta wave, infants under one year of age were 75% more likely to be admitted than the 5-to-14-year-old age group, and patients with rural zip codes were more than twice as likely as those with urban zip codes to be admitted. Patients with a complex medical condition were ten times as likely to be admitted as those without one (**Fig 2A**). During the Delta wave, the pattern of admission risks was similar, but the adjusted ORs were slightly larger. This time, children under one year were >3x as likely to be admitted as children from the 5–14-year-old age group. In addition, children from rural zip codes were >3x as likely to be admitted as those from urban zip codes, while children with a complex care code were 14x as likely to be admitted (**Fig 2B**). During the Omicron wave, children under one-year-old were >2x as likely to be admitted as 5-to-14-year olds, whereas children from rural zip codes were almost 2.9x as likely to be admitted as those from urban zip codes. Finally, those with a complex medical condition had nearly a twenty-fold increase in odds for admission compared to those without (**Fig 2C**).

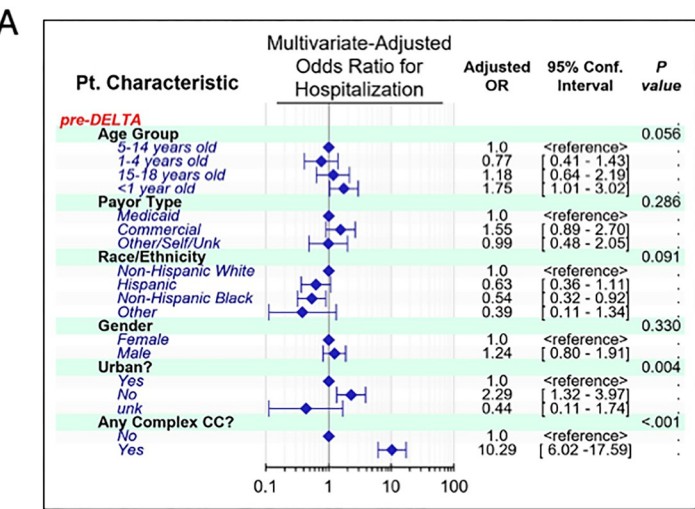

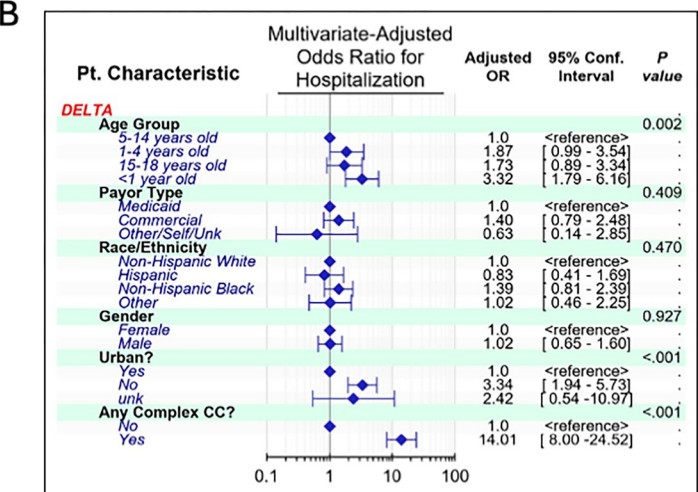

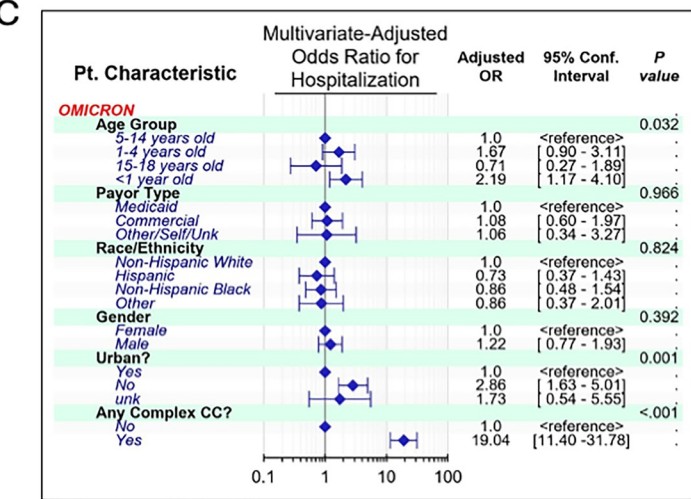

**Fig 2. Forest plots comparing demographic factors within SARS-CoV-2 variant waves.** Children less than one year of age and rural children were more likely to be admitted during the pre-Delta (A), Delta (B), or Omicron (C) waves. Confidence intervals that do not cross 1 on the x-axis are considered significant, with higher numbers associated with increased chances for hospitalization and lower numbers with decreased chances for hospitalization.

## Discussion

This study compared the demographic and clinical characteristics of children presenting with SARS-CoV-2 infections in a large, freestanding pediatric hospital in the Southeastern United States during three variant waves of SARS-COV-2. When comparing those seen in the ED versus those admitted to the hospital, we found subpopulations more likely to be admitted. Throughout all waves, we noted that infants less than one year of age, patients from rural zip codes, and patients with complex medical conditions had significantly increased hospital admission likelihood. These findings suggest a propensity for admissions in those who were younger, rural, and more complex. While this propensity seems obvious for the more complex patients, it warrants closer investigation for the younger and rural patients.

The study hospital is the primary location for inpatient pediatric care in the state, with nearly 95% of the pediatric-specific acute-care beds and the only pediatric intensive-care unit in the state. Furthermore, it is located in the central portion of the state. It is certainly possible that decreased access to care and barriers such as lack of transportation may have influenced clinicians' decisions to admit to the hospital versus discharge home. It is also possible that vaccine uptake within the rural populations in Arkansas was lower. Vaccines became available to children 6–11 years in November of 2021; however, they were not available for those six months to 5 years during the period studied here. McElfish et al. used a cross-sectional design to study the intentions of Arkansan parents/guardians to vaccinate their children against COVID-19 from July 2020 to July 2021. Her team showed that only 26–28% of the population intended to vaccinate their child when it became available. In addition, a full third of those surveyed who planned to get their child vaccinated for COVID-19 right away still had not vaccinated their child two months after approval [18]. While we could not collect vaccination rates, we suspect that vaccine hesitancy for the older children in Arkansas' rural counties may have played a role in lower vaccine uptake with an increased frequency of infections in these areas.

Infants were more likely than other age groups to be admitted to the hospital during all waves. Although more recent evidence [14] has shown that many neonates with COVID have an asymptomatic or mild illness, infants under 60 days have an increased risk for severe bacterial infection. They are likely to be admitted due to institutional practices and policies because of age, regardless of symptoms or viral testing results. This study did not further delineate ages below one year, so we could not determine if younger infants were hospitalized at different rates than older infants. More work is needed to assess the severity of infections in these neonates leading to hospitalization.

A recent study also utilizing PHIS data showed that total hospital admissions in children with chronic conditions decreased by nearly 20% in the first 15 months of the pandemic compared to the same timeframe three years earlier [19]. A 2021 study using another extensive database showed that children with complex and noncomplex chronic diseases were more likely than children without chronic diseases to be hospitalized or have a severe illness with COVID-19 [20]. Similarly, in our study population, children with chronic complex conditions had statistically significant increases in hospital admission compared to those without in all waves.

A large European study showed a 2.50-fold increase in adjusted risk for hospitalization during the Alpha wave and a more modest 1.32-fold increase in this adjusted risk during the Delta wave compared to Omicron [21]. The authors of the European study postulate that the lower risk of hospitalization with the later waves may have been secondary to increased vaccination rates in the pediatric population. We also show pediatric patients were more likely to be admitted to the hospital during the pre-Delta and Delta waves (Adjusted ORs of 2.51 and 2.07,

respectively) than Omicron. However, as mentioned previously, we did not collect vaccination information on our subjects.

Our previous work showed that most demographic characteristics did not significantly differ in the first six months of SARS-CoV-2 infections [14]. In the current study, we see an increase in White non-Hispanic patients and a decrease in the proportion of Hispanic patients, along with no appreciable change among Black patients presenting during the Delta wave. This could be explained secondary to more Hispanics contracting COVID-19 in earlier waves in Arkansas [22] and developing antibodies to future infections. A subsequent increase of Hispanic subjects in our study were infected during the Omicron wave. We suspect those with previous antibodies to other variants were less protected from the Omicron variant, given the mutations in this COVID-19 strain [23–25].

This study has several limitations, including its use of retrospective data from an administrative database. This data relies on billing and coding information and is subject to data reliability problems, including misclassification of exposure and outcomes [26]. Further, our data from a single state may not be generalizable to other states. Our study did not include vaccination status, which may have impacted those patients who were ill enough to present to the emergency department or be admitted to the hospital. Importantly, vaccines were not widely available for children in Arkansas until May 2021 (Adolescents 12–15) (Emergency Use Authorization [EUA]), October 2021 (Children 5–12 years) (EUA), and/or June 2022 (6 months to 5 years) (EUA). In Arkansas, vaccine uptake in children lagged considerably, with only 20–30% of children in metropolitan and 5–15% of rural areas having at least one vaccine by January 2022 [27]. For these reasons, vaccine uptake remains an important public health issue, but may be less important in our population because they were less likely to be vaccinated and can be compared as such. Because race and ethnicity are social constructs, the authors are careful not to draw biological conclusions regarding these data. More prospective studies are needed to evaluate the differences between those hospitalized in our study and those assessed in the emergency department and sent home. Finally, there are likely many other associations that could affect hospitalizations in our patients, including socioeconomic status, the severity of illness, availability of hospital beds, attending bias, etc. Using the PHIS data, we could only account for the variables identified here. Future studies are needed to characterize other associations.

A significant strength of this study is clear definitions of variant waves based on molecular sequencing data from our state, allowing visualization of the impact of each variant on resource utilization and outcomes. Although each participant in this study did not have sequencing data performed, we do have the generic sequencing data associated with our state at the time when each child was admitted. This study was conducted at the only children's hospital in the state, with a large catchment area. Most pediatric COVID patients were likely included in the study, especially if admitted to the hospital. Using an extensive pediatric-specific administrative database offered a large sample size with institutional and geographical representation.

## Conclusions

Children diagnosed with SARS-CoV-2 during the pre-Delta or Delta variants were more likely to be hospitalized than those diagnosed during the Omicron wave. Younger children were more likely to be hospitalized regardless of the wave. The average age of hospitalized patients was lower for the Omicron variant than for Delta and pre-Delta variants. Patients with acute SARS-CoV-2 infection lived in rural areas were more likely to be hospitalized than urban patients. While more studies are required to evaluate the reason for these findings, we suspect

a lower vaccination status and a more considerable distance from medical care could have influenced higher hospitalization rates.

## Supporting information

**S1 Table. As part of the initial methods associated with this study, month-long transition periods were established between waves and after Omicron to ensure that a variant of concern would be present in greater numbers and that there would be less overlap between each variant.** For this reason, we wanted to evaluate the demographic factors associated with these patients.
(DOCX)

**S2 Table. As part of the initial methods associated with this study, month-long transition periods were established between waves and after Omicron to ensure that a variant of concern would be present in greater numbers and that there would be less overlap between each variant.** For this reason, we wanted to evaluate the disease characteristics associated with these patients.
(DOCX)

## Author Contributions

**Conceptualization:** Joshua L. Kennedy.

**Data curation:** Rebecca M. Cantu, Sara C. Sanders, Grace A. Turner, Ashton Ingold, Susanna Hartzell, Dana Frederick, Uday K. Chalwadi, Joshua L. Kennedy.

**Formal analysis:** Rebecca M. Cantu, Grace A. Turner, Uday K. Chalwadi, Eric R. Siegel, Joshua L. Kennedy.

**Funding acquisition:** Joshua L. Kennedy.

**Investigation:** Rebecca M. Cantu, Sara C. Sanders, Joshua L. Kennedy.

**Methodology:** Rebecca M. Cantu, Sara C. Sanders, Eric R. Siegel, Joshua L. Kennedy.

**Project administration:** Rebecca M. Cantu, Suzanne House, Joshua L. Kennedy.

**Resources:** Jessica N. Snowden, Joshua L. Kennedy.

**Supervision:** Rebecca M. Cantu, Jessica N. Snowden, Eric R. Siegel, Joshua L. Kennedy.

**Validation:** Rebecca M. Cantu, Eric R. Siegel, Joshua L. Kennedy.

**Writing – original draft:** Rebecca M. Cantu, Sara C. Sanders, Grace A. Turner, Joshua L. Kennedy.

**Writing – review & editing:** Rebecca M. Cantu, Sara C. Sanders, Grace A. Turner, Jessica N. Snowden, Ashton Ingold, Susanna Hartzell, Suzanne House, Dana Frederick, Uday K. Chalwadi, Eric R. Siegel, Joshua L. Kennedy.

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
