## [Decision Letter · Decision Letter 0]

13 Mar 2024

PONE-D-23-33238Younger and Rural Children are More Likely to be Hospitalized for SARS-CoV-2 Infections.PLOS ONE

Dear Dr. Kennedy,

Thank you for submitting your manuscript to PLOS ONE. After careful consideration, we feel that it has merit but does not fully meet PLOS ONE’s publication criteria as it currently stands. Therefore, we invite you to submit a revised version of the manuscript that addresses the points raised during the review process.

We look forward to receiving your revised manuscript.

Kind regards,

Satyaki Roy, Ph.D.

Academic Editor

PLOS ONE

Journal Requirements:

3. Thank you for stating the following financial disclosure: "Center for Translational Pediatric Research (NIH/NIGMS P20GM121293)

The Translational Research Institute (NIH/NCATS UL1TR003107)"

4. Thank you for stating the following in the Acknowledgments Section of your manuscript: "This work was supported by the Center for Translational Pediatric Research (NIH/NIGMSP20GM121293) at Arkansas Children’s Hospital and by the Translational Research Institute (NIH/NCATS UL1TR003107) at the University of Arkansas for Medical Sciences"

 "Center for Translational Pediatric Research (NIH/NIGMS P20GM121293)

The Translational Research Institute (NIH/NCATS UL1TR003107)"

" Dr. Kennedy has been a consultant for Genentech for asthma. Funding organizations were not involved in the design or conduct of the study, nor were they engaged in the collection, analysis, or interpretation of the data, nor were they involved in the preparation, editing, or censuring of the manuscript."

6. We note that you have indicated that there are restrictions to data sharing for this study. For studies involving human research participant data or other sensitive data, we encourage authors to share de-identified or anonymized data. However, when data cannot be publicly shared for ethical reasons, we allow authors to make their data sets available upon request. For information on unacceptable data access restrictions, please see http://journals.plos.org/plosone/s/data-availability#loc-unacceptable-data-access-restrictions. 

7. Please amend the manuscript submission data (via Edit Submission) to include author Suzanne House.

8. We notice that your supplementary tables are included in the manuscript file. Please remove them and upload them with the file type 'Supporting Information'. Please ensure that each Supporting Information file has a legend listed in the manuscript after the references list.

Additional Editor Comments:

The paper presents a useful study on the factors contributing to pediatric cases of SARS-CoV-2.

The authors are encouraged to submit a minor revision that addresses the reviewer’s comments related to missing explanations on the dataset and improvements to the presentation of the results.

Reviewers' comments:

Reviewer's Responses to Questions

**Comments to the Author**

1. Is the manuscript technically sound, and do the data support the conclusions?

Reviewer #1: Yes

2. Has the statistical analysis been performed appropriately and rigorously? 

Reviewer #1: Yes

3. Have the authors made all data underlying the findings in their manuscript fully available?

Reviewer #1: Yes

4. Is the manuscript presented in an intelligible fashion and written in standard English?

Reviewer #1: Yes

5. Review Comments to the Author

Reviewer #1: This study performed a comprehensive retrospective review at a single center and using the Pediatric Hospital Information System (PHIS) on pediatric cases of SARS-CoV-2 infection leading to hospitalization. This study delineated demographic information and odds ratio for hospitalization, with different subgroup analyses (such as age group, race and gender). Overall, these comprehensive review and data analysis were performed well. There are some minor suggestions that can further improve the manuscript prior to publication.

1. In line 41, the manuscript stated the children that presented to ACH between March 1, 2020, and April 28, 2022 were included. However, in line 71, the start date of pre-Delta period is stated as May 27, 2020. Furthermore, in line 109, the manuscript stated that cases identified were between May 28, 2020 and April 28, 2022. The authors should explain the reasons of differences in the start dates of case identification.

2. Section ‘Data preprocessing’, a cut-off of 30 days was used to differentiate initial and second infection. It is necessary to explain, preferably with reference(s), to support the use of such a cut-off.

3. In line 118, please consider the change of ‘One hundred and thirty’ to ‘130’ as Arabic numbers were used to describe number of patients in line 117 and 119, and 130 should be a large enough number to be used as an Arabic number to start the sentence.

4. For reference no. 26, it would be consistent to change ‘Accessed 8/2023’ to ‘Accessed August 2023’ as the same style as references no. 1 and 15.

5. In table 2, the p-value for univariate analyses in the gender part is 0.0398, which should be considered significant according to the statistical analysis stated. However, the p-value was not bolded. Was this because the significance was adjusted for multiple comparisons? Please explain.

6. Figure 2 can be improved by explaining the x-axis and scale in the figure legend.

7. "Sex" is considered to be male/female as determined at birth, whereas gender is the male/female by preference and chosen. For this cohort, the use of "sex" seems more applicable. Additionally, line 58 states the PHIS Database uses "sex" as well. Therefore, this seems to be a more applicable term for this study. If indeed this is the case, please change throughout the manuscript.

6. PLOS authors have the option to publish the peer review history of their article (what does this mean?). If published, this will include your full peer review and any attached files.

Reviewer #1: **Yes: **Jaime S Rosa Duque

---

## [Author Response · Author response to Decision Letter 0]

12 Jun 2024

Editorial Review.

This has been completed.

This has been completed.

3. Thank you for stating the following financial disclosure: "Center for Translational Pediatric Research (NIH/NIGMS P20GM121293)

The Translational Research Institute (NIH/NCATS UL1TR003107)"

This has been completed and can be found in our cover letter and on the title page.

4. Thank you for stating the following in the Acknowledgments Section of your manuscript: "This work was supported by the Center for Translational Pediatric Research (NIH/NIGMSP20GM121293) at Arkansas Children’s Hospital and by the Translational Research Institute (NIH/NCATS UL1TR003107) at the University of Arkansas for Medical Sciences"

 "Center for Translational Pediatric Research (NIH/NIGMS P20GM121293)

The Translational Research Institute (NIH/NCATS UL1TR003107)" Please include your amended statements within your cover letter; we will change the online submission form on your behalf.

This has been completed. In fact, the acknowledgments section has been removed completely.

" Dr. Kennedy has been a consultant for Genentech for asthma. Funding organizations were not involved in the design or conduct of the study, nor were they engaged in the collection, analysis, or interpretation of the data, nor were they involved in the preparation, editing, or censuring of the manuscript." Please confirm that this does not alter your adherence to all PLOS ONE policies on sharing data and materials, by including the following statement: ""This does not alter our adherence to PLOS ONE policies on sharing data and materials.” (as detailed online in our guide for authors http://journals.plos.org/plosone/s/competing-interests). If there are restrictions on sharing of data and/or materials, please state these. Please note that we cannot proceed with consideration of your article until this information has been declared. Please include your updated Competing Interests statement in your cover letter; we will change the online submission form on your behalf.

This has been completed and is included on the title page and in the cover letter as requested.

6. We note that you have indicated that there are restrictions to data sharing for this study. For studies involving human research participant data or other sensitive data, we encourage authors to share de-identified or anonymized data. However, when data cannot be publicly shared for ethical reasons, we allow authors to make their data sets available upon request. For information on unacceptable data access restrictions, please see http://journals.plos.org/plosone/s/data-availability#loc-unacceptable-data-access-restrictions. 

This has been completed.

This has been completed.

Data Availability Statement: Because of the confidential nature of some of the data and our institutional review board (IRB) restrictions, data will only be provided upon request and with an approved IRB protocol. Data requests can be sent to Dr. Rebecca Cantu (corresponding author).

This has been completed. See above.

7. Please amend the manuscript submission data (via Edit Submission) to include author Suzanne House.

This has been completed.

8. We notice that your supplementary tables are included in the manuscript file. Please remove them and upload them with the file type 'Supporting Information'. Please ensure that each Supporting Information file has a legend listed in the manuscript after the references list.

This has been completed.

This has been completed. Upon my review, I do not see any references that have been retracted. We have changed reference 27 per the reviewer’s concerns (see below). We have also altered reference 15 as the full author was not evident. Finally, we added references 14 and 17 because of the reviewer’s concerns about 30-day infection intervals for secondary infections.

Reviewer #1. 

This study performed a comprehensive retrospective review at a single center and using the Pediatric Hospital Information System (PHIS) on pediatric cases of SARS-CoV-2 infection leading to hospitalization. This study delineated demographic information and odds ratio for hospitalization, with different subgroup analyses (such as age group, race and gender). Overall, these comprehensive review and data analysis were performed well. There are some minor suggestions that can further improve the manuscript prior to publication.

1. In line 41, the manuscript stated the children that presented to ACH between March 1, 2020, and April 28, 2022 were included. However, in line 71, the start date of pre-Delta period is stated as May 27, 2020. Furthermore, in line 109, the manuscript stated that cases identified were between May 28, 2020 and April 28, 2022. The authors should explain the reasons of differences in the start dates of case identification.

We appreciate the reviewer’s concerns. After review of our data, we realized our mistake. The start of the data collection for the PHIS dataset was May 27, 2020. We have made changes to the manuscript to reflect this.

2. Section ‘Data preprocessing’, a cut-off of 30 days was used to differentiate initial and second infection. It is necessary to explain, preferably with reference(s), to support the use of such a cut-off.

We appreciate the reviewer's comments. We utilized the 30-day cut-off for this manuscript because this is the time period that the Centers for Medicare and Medicaid Services considers a readmission for hospital billing purposes. We also used this cut-off in our previous paper (now cited at this point). We recognize that the CDC defines reinfection after 90 days of the first exposure. However, many of these patients were admitted not on their first positive but on their second. CDC also says that the evolving variants increase your risk of reinfection, which can happen as early as “several weeks after a previous infection, although this is rare.” For these reasons, we chose to include data at ≥30 days.

3. In line 118, please consider the change of ‘One hundred and thirty’ to ‘130’ as Arabic numbers were used to describe number of patients in line 117 and 119, and 130 should be a large enough number to be used as an Arabic number to start the sentence.

This has been changed as suggested by the reviewer.

4. For reference no. 26, it would be consistent to change ‘Accessed 8/2023’ to ‘Accessed August 2023’ as the same style as references no. 1 and 15.

This has now been changed as suggested by the reviewer.

5. In table 2, the p-value for univariate analyses in the gender part is 0.0398, which should be considered significant according to the statistical analysis stated. However, the p-value was not bolded. Was this because the significance was adjusted for multiple comparisons? Please explain.

We have now placed this p-value in bold as suggested by the reviewer.

6. Figure 2 can be improved by explaining the x-axis and scale in the figure legend.

We have now made the suggested changes to the figure legend.

7. "Sex" is considered to be male/female as determined at birth, whereas gender is the male/female by preference and chosen. For this cohort, the use of "sex" seems more applicable. Additionally, line 58 states the PHIS Database uses "sex" as well. Therefore, this seems to be a more applicable term for this study. If indeed this is the case, please change throughout the manuscript.

We have now made these changes (lines 101 and 151, Tables 1 and 2, and supplemental Table 1) as suggested by the reviewer.

---

## [Decision Letter · Decision Letter 1]

19 Jul 2024

Younger and Rural Children are More Likely to be Hospitalized for SARS-CoV-2 Infections.

PONE-D-23-33238R1

Dear Dr. Kennedy,

We’re pleased to inform you that your manuscript has been judged scientifically suitable for publication and will be formally accepted for publication once it meets all outstanding technical requirements.

Kind regards,

Satyaki Roy, Ph.D.

Academic Editor

PLOS ONE

Additional Editor Comments (optional):

Reviewers' comments:

Reviewer's Responses to Questions

**Comments to the Author**

1. If the authors have adequately addressed your comments raised in a previous round of review and you feel that this manuscript is now acceptable for publication, you may indicate that here to bypass the “Comments to the Author” section, enter your conflict of interest statement in the “Confidential to Editor” section, and submit your "Accept" recommendation.

Reviewer #1: All comments have been addressed

2. Is the manuscript technically sound, and do the data support the conclusions?

Reviewer #1: Yes

3. Has the statistical analysis been performed appropriately and rigorously? 

Reviewer #1: Yes

4. Have the authors made all data underlying the findings in their manuscript fully available?

Reviewer #1: Yes

5. Is the manuscript presented in an intelligible fashion and written in standard English?

Reviewer #1: Yes

6. Review Comments to the Author

Reviewer #1: This study contributes more data to the literature on SARS-CoV-2 infection leading to hospitalization using the Pediatric Hospital Information System. The authors have addressed the prior comments adequately.

7. PLOS authors have the option to publish the peer review history of their article (what does this mean?). If published, this will include your full peer review and any attached files.

Reviewer #1: **Yes: **Jaime S Rosa Duque

---

## [Editor Report · Acceptance letter]

12 Aug 2024

PONE-D-23-33238R1 

PLOS ONE

Dear Dr. Kennedy, 

I'm pleased to inform you that your manuscript has been deemed suitable for publication in PLOS ONE. Congratulations! Your manuscript is now being handed over to our production team.

Kind regards, 

on behalf of

Dr. Satyaki Roy 

Academic Editor

PLOS ONE